# Light and Shadows in Newborn Screening for Lysosomal Storage Disorders: Eight Years of Experience in Northeast Italy

**DOI:** 10.3390/ijns10010003

**Published:** 2023-12-25

**Authors:** Vincenza Gragnaniello, Chiara Cazzorla, Daniela Gueraldi, Andrea Puma, Christian Loro, Elena Porcù, Maria Stornaiuolo, Paolo Miglioranza, Leonardo Salviati, Alessandro P. Burlina, Alberto B. Burlina

**Affiliations:** 1Division of Inherited Metabolic Diseases, Department of Diagnostic Services, University Hospital of Padua, 35128 Padua, Italy; vincenza.gragnaniello@aopd.veneto.it (V.G.); chiara.cazzorla@aopd.veneto.it (C.C.); daniela.gueraldi@aopd.veneto.it (D.G.); andrea.puma@aopd.veneto.it (A.P.); christian.loro@aopd.veneto.it (C.L.); elena.porcu@aopd.veneto.it (E.P.); mariastorn@gmail.com (M.S.);; 2Division of Inherited Metabolic Diseases, Department of Women’s and Children’s Health, University of Padua, 35128 Padua, Italy; 3Clinical Genetics Unit, Department of Women’s and Children’s Health, University of Padua, 35128 Padua, Italy; leonardo.salviati@unipd.it; 4Neurology Unit, St Bassiano Hospital, 36061 Bassano del Grappa, Italy; alessandro.burlina@aulss7.veneto.it

**Keywords:** newborn screening, lysosomal storage disease, Pompe disease, mucopolysaccharidosis type I, Gaucher disease, Fabry disease, second-tier test, tandem mass spectrometry, glycosaminoglycans, lysosphingosine

## Abstract

In the last two decades, the development of high-throughput diagnostic methods and the availability of effective treatments have increased the interest in newborn screening for lysosomal storage disorders. However, long-term follow-up experience is needed to clearly identify risks, benefits and challenges. We report our 8-year experience of screening and follow-up on about 250,000 neonates screened for four lysosomal storage diseases (Pompe disease, mucopolysaccharidosis type I, Fabry disease, Gaucher disease), using the enzyme activity assay by tandem mass spectrometry, and biomarker quantification as a second-tier test. Among the 126 positive newborns (0.051%), 51 infants were confirmed as affected (positive predictive value 40%), with an overall incidence of 1:4874. Of these, three patients with infantile-onset Pompe disease, two with neonatal-onset Gaucher disease and four with mucopolysaccharidosis type I were immediately treated. Furthermore, another four Gaucher disease patients needed treatment in the first years of life. Our study demonstrates the feasibility and effectiveness of newborn screening for lysosomal storage diseases. Early diagnosis and treatment allow the achievement of better patient outcomes. Challenges such as false-positive rates, the diagnosis of variants of uncertain significance or late-onset forms and the lack of treatment for neuronopathic forms, should be addressed.

## 1. Introduction

Lysosomal storage disorders (LSDs) are inborn errors of metabolism that include about 70 different inherited diseases [1]. They are caused by the deficiency or absence of specific lysosomal enzyme or transporter activities, resulting in the accumulation of uncatabolized macromolecular substrates within lysosomes [2,3,4]. These LSDs can present at any age, with multisystemic involvement and progressive courses. Enzyme analyses on leukocytes/lymphocytes, biochemical assays demonstrating accumulation and gene analyses can confirm the clinical diagnosis [5,6]. The treatment (enzyme-replacement therapy ERT, hematopoietic stem-cell transplantation HSCT, chaperones and gene therapy) can significantly improve outcomes, but the progressive nature of these diseases makes early diagnosis essential [7,8,9,10].

Unfortunately, the diagnosis is often delayed because of non-specific symptoms that frequently arise when organ damage is already irreversibly severe [11,12,13]. This has motivated the development of newborn screening (NBS) programs in the last two decades that use multiplex analytical techniques [5,14,15], such as fluorometry coupled with digital microfluidics (DMF) and electrospray ionization tandem mass spectrometry (MS/MS), for the simultaneous quantification of several lysosomal enzyme activities [16]. 

In Europe, the first pilot NBS project for LSD was performed in the Piemonte region, in Northwest Italy. Between 2003 and 2005, 37,104 male newborns were screened for Fabry disease (FD) using a fluorometric assay. Twelve positive neonates were identified and confirmed as having a specific *GLA* variant (1:3092 males). Only one patient carried a variant associated with the classic phenotype, demonstrating a high incidence of later-onset forms [17]. A similar study, using a fluorometric assay for FD, was performed in 2008 in Galicia (Spain) on 14,600 newborns. Of these, 106 tested positive and 37 carried a *GLA* variant, indicating a very high overall incidence (1:395). However, only one male patient carried a variant associated with the classic phenotype, 11 (including a female neonate) carried variants of uncertain significance (VUS) and 25 carried polymorphisms [18]. 

After the development of multiplex high-throughput assays (DMF and MS/MS), these methods were used to screen for multiple LSDs simultaneously on deidentified neonatal dried blood spot (DBS). Pilot studies using MS/MS to assay four LSDs (Pompe disease PD, Fabry disease FD, Gaucher disease GD and Niemann Pick A/B disease NPD) were performed in Austria and Hungary on 34,736 and 40,024 anonymous neonatal DBS, respectively. Both found an overall incidence of about 1:2300 [19,20]. Subsequently, a similar study performed in Belgium analyzed around 20,000 newborn samples for PD, FD and mucopolysaccharidosis type I (MPS I), using an MS/MS assay to establish reference ranges. The reference intervals were found to depend on sex and gestational age (higher enzyme activities in female and premature newborns) [21]. A limitation of these studies is the lack of follow-up data. 

A retrospective study conducted in Italy identified a high incidence of LSDs, which constituted the most frequent class of inborn errors of metabolism (overall incidence 1:8275) [22]. In 2012, a small pilot program conducted in the Umbria region of Italy assessed four lysosomal disorders (PD, GD, FD and MPS I) using a fluorometric assay. One patient with GD was found among the 3403 newborns screened [23]. 

In 2015, the Regional Health Administration of Northeast Italy expanded its NBS program to include four LSDs, GD, PD, FD and MPS I, using a multiplex high-throughput MS/MS assay. A second-tier test (2TT) of positive DBS (GD, FD, MPS I) was used to reduce the recall rate. Here, we report our 8-year experience with about 250,000 neonates screened, discuss the role of 2TT and briefly describe the patients’ outcomes summarizing the high and low points of our experience. 

## 2. Materials and Methods

### 2.1. Study Population

The DBSs from 248,616 newborns were consecutively collected from September 2015 to August 2023 at the Regional Center for Expanded Newborn Screening, Padua University Hospital. Informed consent was obtained from parents. 

### 2.2. Determination of Enzyme Activities

The DBSs were assayed for the enzyme activities deficient in PD, GD, FD and MPS I (acid α-glucosidase (GAA) for PD, acid β-glucocerebrosidase (GCase) for GD α-galactosidase A (α-GalA) for FD and acid α-L-iduronidase (IDUA) for MPS I) using a flow-injection-MS/MS analysis, as previously reported [24]. The cut-offs (0.2 MoM) were recalculated monthly to avoid an increase in false positives in winter and false negatives in summer, due to seasonal changes in temperature and humidity during transport. 

### 2.3. Second-Tier Tests 

Second-tier tests were introduced in 2016 for GD and FD and in 2019 for MPS I. Newborns who had initial screening results below the cutoff for GCase, α-GalA or IDUA activity were tested for LysoGb1, LysoGb3 and glycosaminoglycans (GAGs), respectively. 

The DBS lysosphingolipids (LysoGb1, LysoGb3) were measured by LC-MS/MS, as previously described [25]. The GAGs levels were measured by methanolysis, followed by LC-MS/MS, with a method optimized for DBSs based on that developed by Zhang et al. [26].

Samples were collected between 36 and 48 h of life at the same time as expanded newborn screening [27] and sent daily to laboratory. The DBSs were analyzed the same day and the results, including 2TT if available, were ready on following day (the test needs an overnight incubation). For recalling sample, the result was available in 8 days.

### 2.4. Confirmatory Testing

Screen-positive newborns were referred to our Clinical Unit for confirmatory testing, including clinical evaluation, enzyme activities in leukocytes/lymphocytes, substrate quantification (plasma LysoGb1 for GD, LysoGb3 for FD, urinary GAGs for MPS I and urinary Glc4 for PD) and mutation analyses. 

For PD and MPS I, which can require immediate intervention, results were obtained within 24 h for urinary Glc4 and GAGs, respectively. Treatment was started immediately in patients with infantile-onset PD (ERT) and Hurler disease (ERT + HSCT). In patients with the neonatal form of GD, treatment was started after ethics discussion. The other patients were followed and treated only when symptoms appeared. Newborns carrying pseudodeficiency variants (changes in the gene sequence that result in reduced activity in vitro, but normal activity in vivo) were dismissed. Family members were counseled and offered testing and medical evaluations.

## 3. Results

### 3.1. Screening Outcomes

Of the 248,616 screened newborns (119,512 males, 129,104 females), 126 positive cases were referred to our Clinical Unit for confirmatory testing (0.05%). Screening results and patients with confirmed diagnoses are reported in Table 1.

#### 3.1.1. Mucopolysaccharidosis Type I

For MPS I, 29/248,616 newborns (0.012%) had low IDUA activity in DBS (range 0.1 to 2.2 µmol/L/h). Of these, 4/29 were confirmed as being affected due to low IDUA activity in the lymphocytes, elevated GAGs in the urine and two pathogenic alleles on genetic analysis. The overall MPS I incidence was 1:62,154 births. 

During the first 3 years of screening (2015–2018, 112,446 newborns), when we only tested IDUA activity in DBS, the recall rate was 0.024% (27 newborns). In total, 25/27 newborns were false positive on confirmatory testing, presenting normal urinary GAGs and genetic variants known to be pseudodeficency alleles (21 newborns, 18 of African origin) or variants of unknown significance (VUS; *n* = 2), or were carriers (*n* = 2) (false-positive rate (FPR) 0.022%, positive predictive value (PPV) 7.4%). The residual enzyme activities in these newborns (0.10–2.21 nmol/L/h) overlapped with those of two true-positive MPS I patients (0.12–0.22 µmol/L/h).

Since 2019 (136,170 newborns), GAG analysis in DBS has been developed as a 2TT. The recall rate during this period was 0.015% (two patients) and no false positives were referred to the clinic (predictive value 100%) (Table 2). 

The four affected newborns were immediately investigated and treated (Table 3).

The first patient, a female of African origin (Morocco), had the intermediate Hurler/Scheie phenotype. She began ERT with laronidase (100 U/kg/weekly) one month after birth with good clinical and biochemical responses.

The other three patients (among whom pt. 2 and pt. 4 were siblings), were of Italian origin, and affected by Hurler syndrome. The use of ERT with laronidase (100 U/kg/weekly) was started early. The GAGs normalized, although in the first months of life, the patients developed mild features of MPS I, such as coarse facial features (*n* = 3), conductive and sensorineural hearing loss (*n* = 2), corneal clouding (*n* = 3) and mild dilatation of the periventricular spaces on brain MRI (*n* = 3). Due to the severity of the phenotype, at 6 months of age, they received an allogenic HSCT. All te patients achieved high donor chimerism and normal IDUA levels. The GAGs remained normal after the discontinuation of ERT, 6 months after the HSCT. Currently, they have no neurologic involvement. 

#### 3.1.2. Pompe Disease

For PD, 48/248,616 newborns (0.019%) needed to be clinically evaluated. Two pathogenic variants of the GAA gene were found in 16/48 newborns, of whom three were infantile-onset (IOPD) and 13 were late-onset (LOPD) patients. The findings in 28 other newborns included a VUS (*n* = 7), a known pseudodeficiency allele or predicted non-pathogenic variant (*n* = 17), or carrier status (*n* = 4). Four newborns were lost to follow-up prior to confirmatory testing due to their relocation outside of the region. The incidence of PD was 1:15,539 (1:10,309 including VUS), of which the IOPD rate was 1:82,872 and the LOPD rate was 1:19,124. The PPV was 52%.

The DBS GAA values in the affected and pseudodeficiency/carrier newborns overlapped (0.45–1.94 µmol/L/h and 1.14–3.22 µmol/L/h, respectively). Only the urinary assay of Glc4 and cardiologic assessment allowed the rapid identification of the IOPD patients. The molecular analyses helped to confirm IOPD and distinguish the LOPD patients from the pseudodeficient and carrier newborns. 

The three IOPD patients were referred between day 3 and day 14 of life. All had increased levels of muscle necrosis markers (CPK 653-1063U/L), uGlc4 (26.5–71.2 mmol/mol crea, nv < 7.4) and hypertrophic cardiomyopathy (LVMI 128–232 g/m^2^). In our laboratory, we assayed the CRIM status on the peripheral blood mononuclear cells, and then confirmed it by using molecular analyses (pt. 1 CRIM pos c.1933G>A, c.2237G>A—p.Asp645Asn, p.Trp746*; pt. 2 CRIM neg c.2560C>T, del exons 4-8—p.Arg854*, del exon 4-8; pt. 3 CRIM neg homozygous c.236_246del—p.Pro79Argfs*13). The patients started ERT (alglucosidase alfa, Genzyme Corp.) between day 5 and day 19 of life, with good responses. To date, all are alive (mean age 4.5 years). Two patients (pt. 1, 5.5 years old, and pt. 3, 3.0 years old) have age-appropriate motor development with no signs of cardiomyopathy and normal biochemical test results, including CPK and uGlc4. However, pt. 2, who was CRIM-negative, developed anti-rhGAA antibodies after 6 months of ERT (max titer 1:102,400), associated with delayed psychomotor development and the persistence of cardiomyopathy [28]. 

Among the patients with LOPD, the typically Caucasian IVS1 variant (IVS1-13T>G) was the most common pathogenic variant, present in 12/13 LOPD cases, six of which were homozygous. Over 8 years (mean follow-up 3.6 years, range 0.5–7), none of the patients carrying a LOPD or VUS variant developed symptoms and none received ERT. 

Among the patients with pseudodeficiency alleles, we found a high incidence of the Asiatic variant c.2065 G>A (p.Glu689Lys), alone (*n* = 1) or in the complex allele c.(1726G>A, 2065G>A) p.(Gly576Ser; Glu689Lys) (*n* = 10). Moreover, five European newborns carried the predicted non-pathogenic variant p.Val222Met. Patients carrying a single variant or pseudodeficiency alleles were dismissed after communication of their molecular assay result.

#### 3.1.3. Fabry Disease

For FD, of the 248,616 newborns screened (119,512 males), 31 newborns (29 males) were referred for confirmatory testing due to low enzyme activity. The α-GalA enzyme activities in the DBSs of the positive patients ranged from 0.45 to 3.45 µmol/L/h (*n* = 29, mean 1.45 µmol/L/h, SD 0.82). Since 2016, LysoGb3 testing was conducted on 24 DBSs with low α-GalA activity. The LysoGb3 values ranged from 0.22 to 8.06 nmol/L (mean 1.38 nmol/L, SD 1.74, nv < 1.45 nmol/L). The values were abnormal in seven newborns. 

All 29 males were confirmed as having low α-GalA activity in lymphocytes and a genetic variant in the *GLA* gene, with an incidence of 1:4121 males. The two female newborns with low DBS α-GalA activity were negative at molecular testing. 

The molecular analyses identified 18 newborns carrying known pathogenic variants (associated with the later-onset form of FD), 10 carrying VUS (including p.Ala143Thr) and one carrying a haplotype considered benign (-10C>T, IVS2-77_81del15, IVS4-16A>G, IVS6-22C>T). The most common pathogenic variant was p.Asn215Ser (*n* = 5). Other variants were associated with ethnic origin. Three unrelated newborns of Asiatic origin carried the splicing variant IVS4+919G>A [12], three unrelated patients of African descent carried the pathogenic variant p.Arg363His and three African patients had arginine substitutions at position 356 (p.Arg356Gln, *n* = 2; and p.Arg356Gly, *n* = 1). Among the patients carrying a VUS, one newborn carrying the p.Ile91Thr variant had abnormal DBS lysoGb3 at birth (8.06 nmol/L), which increased during the follow-up (13.62 nmol/L at 1 yr.), suggesting that this variant may be pathogenic. All the mothers were positive for the same mutation as their children.

All the patients participated in regular follow-up, except for four, whose parents refused additional medical examinations. Because all our patients carried variants associated with the later-onset form or unclassified variants, we decided to follow them every 12 months to avoid overmedicalization. After 8 years of NBS and follow-up, none of our patients (mean age at last visit 4.5 years, range 0.7–7.8 years) had symptoms or signs of disease and none had been treated. All presented a progressive increase in LysoGb3, which was abnormal on the last follow-up in 21/25 patients (mean 1.88 nmol/L, SD 2.2, range 0.35–13.62).

#### 3.1.4. Gaucher Disease 

For GD, positive NBS results were found in 18 neonatal DBS; 16 were referred to our clinic for confirmatory testing, while two infants were lost due to relocation. The diagnosis of GD was confirmed in all the newborns, with an overall incidence of 1:15,539 births. Two newborns affected by the neonatal form of GD were already symptomatic at recall. The other newborns were predicted to have GD1. 

The Gcase activity on the DBS in the confirmed patients ranged from 0.38 to 2.3 uM/h, and the value was not correlated with the disease subtype or severity. Conversely, the LysoGb1 in the DBSs from the patients with GD2 were 7262 nmol/L and 1698 nmol/L (nv < 33.31), respectively, while the mean in the GD1 patients was 130.81 nmol/L (*n* = 14, range 45.55–323). Thus, DBS LysoGb1 at birth can clearly discriminate the disease phenotype. Molecular analyses were available for all 16 patients. Of the two patients with GD2, both of Balkan origin, one was homozygous for p.(His294Gln + Asp448His) and the other compound heterozygous p.Asp448His, p.(His294Gln + Asp448His). Of the 11 genotypes associated with GD1, p.Asn409Ser (*n* = 17) was the most common allele in our cohort, as can be expected for an Italian population. Three patients carried at least one VUS (p.Thr408Met or p.Glu365Lys). 

Except for the two patients who relocated after their diagnosis of GD, most of the patients (*n* = 14) received follow-up evaluations (mean age at last visit 3.8 years, range 0.2–7.0 years). Thus far, 6/16 patients have exhibited clinical manifestations and required therapeutical intervention. The GD2 patients presented at birth with anemia and low platelet counts, petechiae, hepatosplenomegaly and cholestatic hepatitis. They were treated with ERT, beginning at 13 and 4 days of life, and they experienced hematological and visceral improvements. Unfortunately, ERT does not cross the blood–brain barrier, and the first patient presented gradual neurological involvement with delayed psychomotor development [29], while the second patient showed neurological involvement in the first month of life with hypertonus, strabismus, dysphagia, stridor and central apnea. Four GD1 patients developed signs of disease at a mean age of 3 years (range 2–4 years). Three patients developed hepatosplenomegaly and one patient developed bilateral femur osteonecrosis [30]. All started ERT, with good clinical and biochemical outcomes. 

## 4. Discussion

We report our experience after 8 years of LSD screening on about 250,000 newborns, which is the largest study reported in Europe to date. 

Among the 126 newborns who underwent confirmatory tests, we found 51 affected patients carrying pathogenic variants (16 with PD, 13 with GD, 18 males with FD, four with MPS I). Furthermore, 22 asymptomatic newborns were found carrying at least a VUS (seven PD, three GD, 10 FD, two MPS I).

The high recall rate underlines the importance of a 2TT. It can use biomarker quantitation and/or DNA analysis. We chose the use of biomarker quantification as a 2TT analysis for MPSI, GD and FD. The use of molecular genetic analysis as a 2TT still presents several limitations, such as the need for specific informed consent, the identification of unknown mutations or VUS and higher costs.

For MPS I, we found an incidence of 1:62,154, which is higher than the clinically reported value (1:100,000) [31,32,33,34]. In Europe, there have only been a few other experiences with NBS for MPS I, such as the small study in the Umbria region of Italy (3403 newborns) [23] and a pilot project on 64,907 newborns conducted in the Italian regions of Tuscany and Umbria [35]. No true positives were identified in either project, while a total of 15 recalled newborns were false-positive at confirmatory testing. In the USA, NBS for MPS I is included in the Recommended Uniform Screening Panel (RUSP). The incidence ranged from 1:35,509 (Washington State) to 1:219,793 (Illinois) [36], and a high number of pseudodeficiencies is found in programs that do not use 2TT (GAG quantification or DNA sequencing). Conversely, in Taiwan, where the incidence was about 1:73,000, the rate of false positives was very low without 2TT (0.0013% on 294,196 newborns) [37,38]. This was attributed to population homogeneity in Taiwan and a very low frequency of pseudodeficiency. 

Our experience confirmed a high false-positive rate due to pseudodeficiency, which was more prevalent in African and African-American populations [35,39]. Since February 2019, when we added GAGs quantification on DBS as a 2TT [27], our two-test algorithm has reduced false-positive and recall rates, with a PPV of 100% [40]. The outcomes for our patients confirm that early HSCT can change the natural history of the severe disease form, preventing neurological involvement and allowing normal psychomotor development. 

During the study period, one patient was clinically diagnosed at the age of 10 months with Hurler syndrome. The patient was born in a neighboring country and did not undergo neonatal screening for lysosomal diseases. At diagnosis, he presented with severe organ involvement (coarse facies, lumbar hump, joint stiffness, heart valve disease, visceromegaly, umbilical hernia, corneal opacity and severe deafness). This confirms the importance of neonatal screening for MPS I in avoiding diagnostic delay. No screen-negative patients were clinically diagnosed, with a negative predictive value of 100%.

The incidence of PD from our NBS data was 1:10,309 (IOPD 1:82,872; LOPD 1:19,124). The reported clinical prevalence of PD worldwide is 1:40,000 [41], whilst a previous Italian study estimated an incidence of 1:120,743 [22]. The difference between these and our results might be explained by recent immigration from Africa, where there is a higher incidence of consanguinity, but also by the identification of a high number of LOPD patients, as reported in other screening programs. Among 48 newborns with a positive NBS result, three patients had IOPD, 13 were predicted to have LOPD and seven carried a VUS (PPV 52%). 

In IOPD, treatment should be started as soon as possible; delays of a few days can influence outcomes [42,43]. Our IOPD patients (*n* = 3) started ERT very early, between 5 and 19 days of life. Two had good outcomes, while one patient, who was CRIM-negative, developed a non-compactum myocardium and psychomotor delay [28] due to the formation of anti-ERT antibodies [44]. 

In patients with LOPD, ERT is associated with better outcomes when started before irreversible muscle damage occurs [45,46]. The outcome of LOPD is difficult to predict. Most of our patients carried the IVS1 variant, which is the most frequent variant in the Caucasian population (90% of patients) [47], with phenotypic variation in the age of symptom onset from early childhood to late adulthood [48]. Currently, our LOPD patients (mean age 3.4 years, range 0.3–6.9 years) have no clinical signs or symptoms, all have age-appropriate development and none is receiving ERT. 

Our data demonstrate a high incidence of pseudodeficiencies, especially of the Asiatic variant p.Glu689Lys, alone (*n* = 1) or in the complex allele p.(Gly576Ser; Glu689Lys) (*n* = 7). Moreover, five European newborns carried the predicted non-pathogenic variant p.Val222Met [49].

Despite these reservations, NBS for PD is widespread worldwide. In the USA, it was the first LSD to be included in the RUSP, added in 2015. All the reported programs confirmed a higher-than-expected incidence of disease and, in particular, a high number of LOPDs. In Taiwan, the first NBS pilot study of PD was conducted in 2005. The Taiwanese cohort is unique, because almost all the IOPD patients are CRIM-positive (due to the high frequency of the p.Asp645Glu mutation), LOPD cases lack the IVS1 variant, which is common in Caucasian population, and there is a high frequency of the pseudodeficiency allele p.(Gly576Ser; Glu689Lys).

For GD, among 18 newborns with positive NBS, we diagnosed 13 patients with GD1 and two patients with GD2, while three patients carried a VUS (PPV 100%). The overall incidence was 1:16,574, which is three times higher than the clinically estimated rate (1:40,000–1:60,000) [50]. The most frequent allele was p.Asn409Ser (*n* = 17, associated with phenotype GD1); interestingly, it was followed by the p.(His294Gln+Asp448His) allele (*n* = 4), of Balkan origin, which is associated with a GD2/3 phenotype [51]. This was probably due to recent immigration from the Balkans. 

The application of NBS for GD is still controversial, and in the USA, it is not included in the RUSP. Only a few states have reported their experience, such as Missouri and Illinois (43,701 and 219,973 newborns, respectively), which found an incidence of about 1:43,000 [52,53]. In New York State (65,605 newborns), a very high incidence was found (1:4374), and this probably reflects New York City’s large Jewish population, in which the p.Asn409Ser allele is frequent [54]. Several NBS programs for GD have been conducted in East Asia, where, due to the presence of different ethnic backgrounds, a lower incidence of disease was reported (1:73,743 to 1:103,134), and most of the patients suffered from neuronopathic forms (GD2/3), for which few therapeutic options are available [37,55,56,57,58].

There are few data on the measurement of LysoGb1 in the neonatal period. Its use as a 2TT appears promising. In our population, all the neonates with increased LysoGb1 were confirmed as true-positive, with a PPV of 100%. The presence of LysoGb1 at birth was correlated with disease burden, in agreement with the genotype, as patients harboring neuropathic alleles returned significantly higher plasma concentrations than GD patients with non-neuropathic disease. Interestingly, six out of sixteen patients, including four GD1 patients, started ERT in their first years of life due to clinical manifestations (hepatosplenomegaly, osteonecrosis). This underlines the importance of NBS for GD to prevent irreversible complications and avoid diagnostic delays. 

In our study, FD was identified and confirmed in 28 males, while one newborn carried a benign variant. The two female newborns were found to be false positives (overall PPV 90%). The overall incidence of α-GalA deficiency was 1:8879 (1:4121 males), which is similar to previous reports from NBS programs [59], but higher than the clinically estimated incidence (1:40,000) [60], probably due to the recognition of previously undiagnosed later-onset forms of FD. Ten patients carried unclassified variants. It is difficult to predict their pathogenicity because FD may occur later in life. Clinically, our patients presented no symptoms or signs on their last visit (at 0.7–7.8 years of age), although they showed a progressive increase in LysoGb3. All the mothers carried the same mutation as their children. 

In the USA, FD is not included in the RUSP, but local laws promoted by NBS advocates and parents allowed its implementation in NBS in several states. These programs confirmed the high incidence of the disease (1:1790 to 1:15,558), but also the high number of false positives [59]. Conversely, in Taiwan, NBS for FD was started in 2006. The disease prevalence was very high due the diffusion of the IV4+919G>A variant (82%, 1:1600 males), which is associated with cardiac involvement [61]. 

We tested the utility of LysoGb3 as a 2TT [62]. Only 7/29 patients had abnormal DBS LysoGb3 at birth, indicating that a normal result cannot exclude FD. Thus, it is not a reliable 2TT in the neonatal period. However, a recent report of two brothers with classic FD (genotype c. 370–2 A>G) showed that LysoGb3 was elevated in the first days of life and that it increased significantly during infancy in these patients [63]; therefore, LysoGb3 values may contribute to the complete identification of the phenotype. 

We did not detect any heterozygotes among the 129,104 females screened, which confirms that the current enzyme-based NBS approach misses most female carriers due to X-chromosome inactivation. Some authors suggest first-tier screening with *GLA* gene sequencing in female newborns. This method has been applied in Taiwan, where 21 variants account for approximately 98% of the variants [64]. In Italy and throughout Europe, mutational heterogeneity hampers the use of molecular analysis for high-throughput screening, which may also increase the number of VUS. 

## 5. Limitations 

The use of NBS for LSDs (MPS I, PD, GD and FD) has several limitations:A high number of false positives and pseudodeficiencies, which affect families and healthcare systems. Our results demonstrate that biochemical 2TTs are effective in reducing the recall rate for MPS I (GAG quantification) and GD (LysoGb1). In FD, the use of LysoGb3 in the neonatal period is debated [59]. The development of a 2TT is important for all the screened diseases. The approach to reducing the false-positive rate for PD includes both biochemical (creatine/creatinine over GAA ratio by MS/MS) [65] and/or molecular 2TTs [66,67,68] and postanalytical tools, e.g., CLIR [69,70].The identification of an increasing number of VUSs, especially for PD and FD. Some of these patients may never develop symptoms; however, they require ongoing monitoring, which causes anxiety for families and costs for healthcare systems. Future improvements in the management of these patients are likely to arise from a better understanding of the genotype/phenotype correlation and the natural history of the disease.The use of NBS detects newborns across the full phenotypic spectrum of severity, although not all subtypes benefit equally from early detection. Central nervous system damage from neuropathic LSDs, such as GD2, cannot be treated currently [71]. Despite the benefit of informing genetic counseling for parents, an untreatable disease may not fit the criteria for NBS. Potential new therapies for the central nervous system are under development.Ethical issues due to the high incidence of late-onset forms. For these disorders, the onset is often unpredictable and some “patients-in-waiting” may never develop symptoms [72], resulting in unnecessary anxiety and medical intervention, but also the potential treatment of these patients as “vulnerable children” [73]. Challenges remain in identifying better biomarkers for phenotype prediction and the best tailored strategy for the follow-up, management and treatment of these patients.

In conclusion, our experience showed that NBS for these four LSDs is feasible and extendable to larger populations. The incidence of screened LSDs appears to be higher than the rate clinically estimated in our country, but compatible with other screening studies. The possibility of including 2TTs for three of these diseases has changed the false-positive rate. 

Our experience also outlines the importance of early diagnosis in improving disease outcomes when treatments are available. 

The remaining challenges include determining the best management of infants with late-onset phenotypes and those carrying VUSs. The long-term follow-up of these patients is an emerging problem. The sharing of data through international databases will provide valuable information on genotype–phenotype correlations, the natural history, the role of biomarkers and the impact of early treatment, all of which will pave the way to the optimized management of these patients. 

## Figures and Tables

**Table 1 IJNS-10-00003-t001:** Newborn-screening results and confirmed diagnoses.

LSD	GD	PD	FD	MPS I	TOT
Total screened	248,616				
Positive NBS	18	48	31	29	126
Patients with confirmed disorder	13	16	18	4	51
Pseudodeficiency	0	17	0	21	38
VUS	3	7	10	2	22
Carrier status	0	4	/	2	6
Benign variant	0	0	1	0	1
No variant	0	0	2 *	0	2
Lost before confirmatory testing	2	4	0	0	6
PPV	100%	52%	90%	86%	63%
Incidence	1:15,539	1:10,809	1:8879	1:62,154	1:3406

* Both were females. Pseudodeficiency: changes in the gene sequence that result in reduced activity in vitro, but normal activity in vivo. VUS: variant of uncertain significance or unclassified.

**Table 2 IJNS-10-00003-t002:** MPS I NBS results before and after the use of the 2TT.

	2015–2018 (Only IDUA Assay)	2019–2023 (DBS GAGs as 2TT)
Total screened	112,446	136,170
Positive NBS	27	2
Patients with confirmed disorder	2	2
Pseudodeficiency	21	0
VUS	2	0
Carrier status	2	0
PPV	7.4%	100%

**Table 3 IJNS-10-00003-t003:** Clinical and biochemical characteristics of MPS I patients.

	Pt. 1 (Female)	Pt. 2 (Female)	Pt. 3 (Male)	Pt. 4 (Female)
**Diagnosis**				
IDUA activity on DBS	0.22 µmol/L/h	0.17 µmol/L/h	0.17 µmol/L/h	0.12 µmol/L/h
GAGs on DBS HS nv < 3.2 µg/mL; DS nv < 2.7 µg/mL	NA	NA	HS 6.3 µg/mL, DS 33 µg/mL	HS 7.8 µg/mL, DS 41 µg/mL
uGAGs HS nv < 4.6 mg/mmol crea; DS nv < 38.1 mg/mmol crea	HS 148.9 mg/mmol crea, DS 172 mg/mmol crea	HS 121.9 mg/mmol crea and DS 80 mg/mmol crea	HS 60.9 mg/mmol crea and DS 58 mg/mmol crea	HS 274 mg/mmol crea and DS 287.7 mg/mmol crea
Molecular analysis (DNA)	homozygous c.1598C>G	homozygous p.Tyr201*	c.603C>G + c.-14_10del	homozygous c.46_57del12
Protein	homozygous p.Pro533Arg	homozygous p.Tyr201*	pTyr201* + intronic variant	homozygous p.Tyr201*
Phenotype	Hurler/Scheie	Hurler	Hurler	Hurler
**Treatment**				
ERT (laronidase 100 U/kg) (age)	30 days	45 days	19 days	9 days
HSCT (age)	no	6 months	6 months	6 months
**Last follow-up**				
Age	6.5 years old	6 years old	1.5 years	1 year
Clinical features	diffuse corneal clouding	mild coarse facial features, deformity of lumbar vertebral bodies, diffuse corneal clouding.	Mild deformity of vertebral bodies, diffuse corneal clouding.	deformity of thoracic and lumbar vertebral bodies, diffuse corneal clouding.
Cognitive function	WPPSI III: 93	WPPSI III: 121	Bayley 3: 75	Bayley 3: 100
Chimerism	/	94% donor	86% donor	100% donor
IDUA nv 1.9–15 µmol/L/h	/	6.49 µmol/L/h	3.49 µmol/L/h	12.60 µmol/L/h
uGAGs	HS 9.2 mg/mmol crea, DS 15.4 mg/mmol crea	HS 2.6 mg/mmol crea, DS 5.5 mg/mmol crea	HS 3.9 mg/mol crea, DS 12 mg/mmol crea	HS 8.4 mg/mmol crea, DS 13.2 mg/mmol crea

NA: not available, nv: normal value; HS: heparan sulfate, DS: dermatan sulfate, crea: creatinine.

## Data Availability

Data available on request due to privacy/ethical restrictions.

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
