# Peer review of "Light and Shadows in Newborn Screening for Lysosomal Storage Disorders: Eight Years of Experience in Northeast Italy"

_2409-515X, 2023, doi:10.3390/ijns10010003_

Round 1

Reviewer 1 Report

Comments and Suggestions for Authors

The authors of this manuscript describe their experience with LSD screening, after screening about 250,000 babies in the last 8 years. The information is clearly presented and will of interest to other newborn screening around the world. Some interest comparisons are done to the US and Taiwan programs and I appreciated the information on challenges/limitations for screening of those disorders. I have only a few minor suggested edits:

1) I found the information regarding the effectiveness of 2nd tier-screening very interesting along with the false positives rates, PPV, NPV etc. All that information is spread all over the narrative and difficult to track down. I would suggest that the authors modify  table 2 to summarize that information. I would put in columns the 4 LSD disorders and then in rows the information currently depicted, along with info on first tier positive, second tier positives (I understand that the N will be a lower number than the total screened (248,616) as 2TM implemented later on), first and second-tier false positives, PPV, NPV etc. I understand that not all disorders had 2TM. Again this is just a suggestion to try to help summarize that very important info.

2) Line 235: Change nmol/l to nmol/L.

3) Line 364: Change lysoGB3 to LysoGB3

3) Line 364-368: I found interesting the section of the utility of LysoGB3. It is unclear how the authors are currently using that measurement. Sine a normal LysoGb3 result cannot exclude FD, then I assume that assay can not really used as a 2TT and only maybe useful for its potential contribution to the complete identification of the phenotype by periodically testing for it during infancy that goes beyond the NBS needs.

Author Response

The authors of this manuscript describe their experience with LSD screening, after screening about 250,000 babies in the last 8 years. The information is clearly presented and will of interest to other newborn screening around the world. Some interest comparisons are done to the US and Taiwan programs and I appreciated the information on challenges/limitations for screening of those disorders. I have only a few minor suggested edits:

1) I found the information regarding the effectiveness of 2nd tier-screening very interesting along with the false positives rates, PPV, NPV etc. All that information is spread all over the narrative and difficult to track down. I would suggest that the authors modify table 2 to summarize that information. I would put in columns the 4 LSD disorders and then in rows the information currently depicted, along with info on first tier positive, second tier positives (I understand that the N will be a lower number than the total screened (248,616) as 2TM implemented later on), first and second-tier false positives, PPV, NPV etc. I understand that not all disorders had 2TM. Again this is just a suggestion to try to help summarize that very important info.

R: Thank you for the important comment, that improve our manuscript. We have rewritten Table 1 as you suggested. Moreover, due to the dramatic change observed in MPS I after the introduction of 2TT, we have added a specific table for MPS I (table 2).

2) Line 235: Change nmol/l to nmol/L.

R: Thank you for the comment. We have changed as suggested.

3) Line 364: Change lysoGB3 to LysoGB3

R: Thank you for the comment. We have changed as suggested.

3) Line 364-368: I found interesting the section of the utility of LysoGB3. It is unclear how the authors are currently using that measurement. Sine a normal LysoGb3 result cannot exclude FD, then I assume that assay can not really used as a 2TT and only maybe useful for its potential contribution to the complete identification of the phenotype by periodically testing for it during infancy that goes beyond the NBS needs.

R: Thank you for the valuable comment. Your interpretation is correct. We added in the text that “It (LysoGb3) is not a reliable 2TT in neonatal period”.

Reviewer 2 Report

Comments and Suggestions for Authors

This article by Vincenza Gragnaniello and colleagues well recapitulate 8 years of newborn screening for lysosomal storage disorders in Northeast Italy, with the positive aspects and the challenges still to be overcome. The work is well done and of considerable importance. I just have a few suggestions for authors that I think will make the manuscript easier to read.

1.       Line 43. The number of lysosomal diseases is not defined with certainty. Often in the papers the authors indicate 50 and growing, sometimes even 70 (Platt et al. Nat Rev Dis Primers 4, 27 (2018)). In this case authors report 60 LSDs but the cited reference reports 50. Please correct with an appropriate reference that agrees with what is indicated in the text.

2.       Line 48. I would change "diagnosis" with "clinical diagnosis".

3.       Line 51. I would add some references, including more recent ones (see as an example: Furlano et al. Case Rep Genet. 2022:3208810; Pjetraj et al. Am J Med Genet A. 2023;191(2):564-569; Yang et al. J Med Genet. 2023;60(5):430-439).

4.       Line 76. The abbreviation MPSI is not used previously in the text. Please specify.

5.       Line 98-100.  For easier reading, please follow the same order as line 98.

6.       Table 3.

a.       Please specify whether the days indicated in the "treatment" section refer to birth or diagnosis.

b.       Are the clinical features referred to the last follow up?

c.       The clinical features in the table do not completely match those in the text

7.       Line 140. The abbreviation PPV is not used previously in the text. Please specify.

8.       Line 148. In the abbreviations also include "nv"

9.       Line 202. Why was LysoGb3 testing only conducted on 24 samples and not 29?

10.   Line 208-219. The reporting of variants for FD is not immediate. There is a risk of confusion between pathogenetic variants and VUS. Please reformulate for greater clarity, possibly also aided by a table if the authors deem it more useful.

11.   Line 261. Authors reported here 13 patients with GD and 18 males with FD. However, in the Results section, they reported 16 GD and 29 FD. Are these differences due to VUS? In the case of GD, 3/16 are reported as VUS, but 14/16 are under follow-up. Please clarify these discrepancies further.

Finally, I would suggest the authors include a list of abbreviations in the paper to make reading easier.

Author Response

This article well recapitulate 8 years of newborn screening for lysosomal storage disorders in Northeast Italy, with the positive aspects and the challenges still to be overcome. The work is well done and of considerable importance. I just have a few suggestions for authors that I think will make the manuscript easier to read.

1. Line 43. The number of lysosomal diseases is not defined with certainty. Often in the papers the authors indicate 50 and growing, sometimes even 70 (Platt et al. Nat Rev Dis Primers 4, 27 (2018)). In this case authors report 60 LSDs but the cited reference reports 50. Please correct with an appropriate reference that agrees with what is indicated in the text.

R: Thank you for the important comment. We have updated the reference as you suggested.

2. Line 48. I would change "diagnosis" with "clinical diagnosis".

R: Thank you for the comment. We have changed the text.

3. Line 51. I would add some references, including more recent ones (see as an example: Furlano et al. Case Rep Genet. 2022:3208810; Pjetraj et al. Am J Med Genet A. 2023;191(2):564-569; Yang et al. J Med Genet. 2023;60(5):430-439).

R: Thank you for the suggestion. We have added the references.

4. Line 76. The abbreviation MPSI is not used previously in the text. Please specify.

R: Thank you for the comment. We have specified the abbreviation.

5. Line 98-100. For easier reading, please follow the same order as line 98.

R: Thank you for your comment. We have modified the order.

6. Table 3.

a. Please specify whether the days indicated in the "treatment" section refer to birth or diagnosis.

R: Thank you for the comment. It is referred to the age of the patients, we have added this data in the table.

b. Are the clinical features referred to the last follow up?

R: Thank you for the comment. We have specified that the clinical features are referred to the last follow up.

c. The clinical features in the table do not completely match those in the text

R: Thank you for the valuable comment. The clinical features in the Table are referred to the last follow up, while those in the text are referred to the baseline (before HSCT). We have better clarified in the manuscript.

7. Line 140. The abbreviation PPV is not used previously in the text. Please specify.

R: Thank you for the comment. We have specified the abbreviation.

8. Line 148. In the abbreviations also include "nv"

R: Thank you for the comment. We have added the abbreviation.

9. Line 202. Why was LysoGb3 testing only conducted on 24 samples and not 29?

R: Thank you for the important comment. LysoGb3 testing was introduced in 2016, as explained in Materials and Methods section. We have specified this data also in the result section.

10. Line 208-219. The reporting of variants for FD is not immediate. There is a risk of confusion between pathogenetic variants and VUS. Please reformulate for greater clarity, possibly also aided by a table if the authors deem it more useful.

R: Thank you for the comment that allow us to improve the manuscript. We have reformulated the sentence as follows: “Molecular analysis identified 18 newborns carrying known pathogenic variants (associated with the later-onset form of FD), 10 carrying VUS (including p.Ala143Thr) and 1 carrying a haplotype considered benign ( 10C>T, IVS2-77_81del15, IVS4-16A>G, IVS6-22C>T).”

11. Line 261. Authors reported here 13 patients with GD and 18 males with FD. However, in the Results section, they reported 16 GD and 29 FD. Are these differences due to VUS? In the case of GD, 3/16 are reported as VUS, but 14/16 are under follow-up. Please clarify these discrepancies further.

R: Thank you for your valuable comment. We have reformulated as follows:Among the 126 newborns who underwent confirmatory tests, we found 51 affected patients, carrying pathogenic variants (16 with PD, 13 with GD, 18 males with FD, 4 with MPS I). Furthermore, 22 asymptomatic newborns were found carrying at least a VUS (7 PD, 3 GD, 10 FD, 2 MPS I).”

Finally, I would suggest the authors include a list of abbreviations in the paper to make reading easier.

R: Thank you for the important suggestion. We have added a list of abbreviations.

Reviewer 3 Report

Comments and Suggestions for Authors

This manuscript brings a significative contribution to the field of expanded newborn screening, focusing on the topic of NBS for LSDs. The experience on the screening of 4 LSDs in approximately 250,000 babies is described in detail, and this represents a useful contribution to the people involved with this topic. I have just a few minor concerns, listed below.

1) A definition of pseudodeficiency should be included in Material and Methods section; perhaps a footnote on Table 2 defining what is pseudodeficiency and what is VUS may be helpful;

2) It was not clear if the genetic testing was performed in the same sample used for the initial enzyme assay or if it was made in a sample collected later; if not performed, the possibility of performing the genetic testing in the first sample without recalling the patient could be included in the discussion;

3) I suggest to include a table indicating the time from collection to first result, the time between first result and collection of second sample, and the time between collection of second sample and final result, indicating the times for each of the 4 conditions; 

4) The severe form of Gaucher disease is sometimes referred as "GD Neonatal Onset" or "GD2"; I suggest to stardardize it and use always GD2.

Author Response

This manuscript brings a significative contribution to the field of expanded newborn screening, focusing on the topic of NBS for LSDs. The experience on the screening of 4 LSDs in approximately 250,000 babies is described in detail, and this represents a useful contribution to the people involved with this topic. I have just a few minor concerns, listed below.

1) A definition of pseudodeficiency should be included in Material and Methods section; perhaps a footnote on Table 2 defining what is pseudodeficiency and what is VUS may be helpful;

R: Thank you for the comment. We have added in Materials and Methods that “Newborns carrying pseudodeficiency variants (changes in the gene sequence that result in reduced activity in vitro, but normal activity in vivo) were dismissed.” and as footnote of table 1: “Pseudodeficiency: changes in the gene sequence that result in reduced activity in vitro, but normal activity in vivo. VUS: variant of uncertain significance or unclassified.”

2) It was not clear if the genetic testing was performed in the same sample used for the initial enzyme assay or if it was made in a sample collected later; if not performed, the possibility of performing the genetic testing in the first sample without recalling the patient could be included in the discussion;

R: Thank you for the important comment. We have added in the discussion that “The high recall rate underlines the importance of a 2TT. It can use biomarker quantitation and/or Dna analysis. We chose the use of biomarkers quantification as 2TT analysis for MPSI, GD and FD. The use of molecular genetic analysis as 2TT still presents several limitations, such as the need of a specific informed consent, the identification of unknown mutations or VUS and higher costs.”

3) I suggest to include a table indicating the time from collection to first result, the time between first result and collection of second sample, and the time between collection of second sample and final result, indicating the times for each of the 4 conditions;

R: Thank you for the important comment, that allow us to better clarify the timing of NBS. We have added in Material and methods that “Samples were collected between 36 and 48 h of life at the same time of expanded newborn screening and sent daily to laboratory. DBS were analyzed the same day and the results, including 2TT if available, ready in following day (the test needs an overnight incubation). For recalling sample, the result is available in 8 days.”

4) The severe form of Gaucher disease is sometimes referred as "GD Neonatal Onset" or "GD2"; I suggest to stardardize it and use always GD2.

R: Thank you for the comment. We have modified as suggested.